# Chemotherapy resistance due to epithelial-to-mesenchymal transition is caused by abnormal lipid metabolic balance

Atsushi Matsumoto[1†], Akihito Inoko[2,3†], Takuya Tanaka[4], Gen-Ichi Konishi[4], Waki Hosoda[5], Takahiro Kojima[6], Koji Ohnishi[2], Junichi Ikenouchi[1*]

[1]Department of Biochemistry, Kyushu University Graduate School of Medical Sciences, Fukuoka, Japan; [2]Department of Pathology, Aichi Medical University School of Medicine, Nagakute, Japan; [3]Division of Cancer Epidemiology and Prevention, Aichi Cancer Center Research Institute, Nagoya, Japan; [4]Department of Chemical Science and Engineering, Institute of Science Tokyo, Meguro-ku, Tokyo, Japan; [5]Department of Pathology and Molecular Diagnostics, Aichi Cancer Center Hospital, Nagoya, Japan; [6]Department of Urology, Aichi Cancer Center Hospital, Nagoya, Japan

**\*For correspondence:**
ikenouchi.junichi.033@m.kyushu-u.ac.jp

[†]These authors contributed equally to this work.

**Competing interest:** The authors declare that no competing interests exist.

## eLife Assessment

This article presents the **fundamental** discovery that lipid metabolic imbalance induced by Snail, an EMT-related transcription factor, contributes to the acquisition of chemoresistance in cancer cells. The evidence, supported by a wide range of methods and adequate quantification, provides a **convincing** mechanistic explanation of how Snail drives ectopic expression of the cholesterol- and drug-efflux transporter ABCA1. This work, which introduces a novel therapeutic concept targeting invasive cancer, will be of broad interest to researchers in cancer biology, lipid metabolism, and cell biology.

**Abstract** Invasive cancer is defined by the loss of epithelial cell traits resulting from the ectopic expression of epithelial–mesenchymal transition (EMT)-related transcription factors such as Snail. Although EMT is known to impart chemoresistance to cancer cells, the precise molecular mechanisms remain elusive. We found that Snail expression confers chemoresistance by upregulating the cholesterol efflux pump ABCA1 as a countermeasure to the excess of cytotoxic free cholesterol relative to its major interaction partner in cellular membranes, sphingomyelin. This imbalance is introduced by the transcriptional repression of enzymes involved in the biosynthesis of sphingomyelin by Snail. Inhibiting esterification of cholesterol, which renders it inert, selectively suppresses growth of a xenograft model of Snail-positive kidney cancer. Our findings offer a new perspective on lipid-targeting strategies for invasive cancer therapy.

## Introduction

Metastatic cancer, characterized by the spread of tumor cells to distant organs, remains a major challenge in cancer treatment due to its high mortality rate. Metastasis involves a series of biological events, such as acquisition of cancer stem cell characteristics, tumor–microenvironment interactions,

alterations in cell metabolism and resistance to chemotherapy, that are mediated by epithelial–mesenchymal transition (EMT) (*Nieto et al., 2016*; *Lambert and Weinberg, 2021*; *Zheng et al., 2015*; *Fischer et al., 2015*). Transcription factors driving EMT (EMT-TFs), including Snail, Slug, and Twist, that are ectopically expressed in invasive cancers were initially thought to promote invasion and metastasis by suppressing the expression of adhesion molecules in cancer cells. However, since E-cadherin-mediated adhesion is essential for cell survival after metastasis (*Padmanaban et al., 2019*), invasive cancer cells do not undergo complete EMT but rather sustain a hybrid state with both epithelial and mesenchymal (hybrid E/M) characteristics. In a mouse model of pancreatic carcinoma and breast cancers where key EMT-TFs were deleted, the suppression of EMT did not alter the appearance of systemic dissemination or metastasis but rather contributed to increased sensitivity to anticancer drugs (*Zheng et al., 2015*; *Fischer et al., 2015*). Therefore, an in-depth understanding of the molecular mechanisms of resistance to anticancer drugs conferred by EMT-TFs is an essential prerequisite for devising effective therapeutic interventions (*Davis et al., 2014*). It was recently reported that expression of a Rho family GTPase, RHOJ, is upregulated in cancer cells undergoing EMT and that its activation enhances the DNA damage response, which enables tumor cells to efficiently repair chemotherapy-induced DNA damage (*Debaugnies et al., 2023*). Here, we investigated EMT-dependent acquisition of resistance to anticancer agents that do not directly cause DNA damage, such as those targeting the ERK signaling pathway, and found that expression of EMT-TFs causes metabolic alterations of sphingomyelin, resulting in ectopic induction of drug efflux transporter expression and conferring chemotherapy resistance to cancer cells.

## Results

### Snail-induced ABCA1 expression confers chemoresistance to hybrid E/M cells

Renal cell carcinoma (RCC) is notorious for the lack of sensitivity to chemotherapy (*Makhov et al., 2018*), but the underlying mechanisms are not elucidated. Nitidine chloride (NC) holds promise as a drug treatment against RCC as it was shown to have a pro-apoptotic effect on RCC and inhibit tumor growth by suppressing the ERK signaling pathway in a xenograft model (*Cui et al., 2020*; *Fang et al., 2014*). However, in human lung adenocarcinoma cells, NC efficacy was inversely correlated with the expression level of the drug transporter ABCA1: cancer cells with downregulated ABCA1 were more sensitive to NC treatment (*Iwasaki et al., 2010*). Therefore, we wondered whether ABCA1 could also confer NC resistance in RCC. We treated five different RCC cell lines with a combination of NC and an ABCA1 inhibitor cyclosporin A (CsA). Antiproliferative effect of NC was significantly enhanced in the presence of CsA for three lines among five, Caki-1, 786-O, and A498 (*Figure 1A*). These cells consistently expressed higher levels of ABCA1 compared to the nonresponsive cells (*Figure 1B and C*), suggesting the acquisition of ABCA1-mediated drug resistance. Notably, ABCA1 expression was positively correlated with expression of the EMT-TF Snail in the RCC cell lines (*Figure 1D and E*). To investigate the clinical relevance of ABCA1 upregulation in cancers, we analyzed *ABCA1* expression in cancer using the UCSC Xena platform. Comparative *ABCA1* expression data of both normal tissue and primary tumors were available for 24 cancer subtypes in the GDC TCGA database. In addition to glioblastoma and head and neck cancer, all three types of renal cancers (clear cell carcinoma [ccRCC], chromophobe carcinoma, and papillary cell carcinoma) exhibited significant *ABCA1* upregulation in tumor (*Figure 1F*, *Figure 1—figure supplement 1A and B*), indicating that ABCA1 upregulation frequently occurs in clinical renal cancers.

To examine whether the enrichment of ABCA1 protein in malignant cancer cells is detectable in clinical cases, we conducted immunohistochemical analysis on surgical specimens taken from patients with primary ccRCC. The findings showed that ABCA1 was upregulated at the lesion site of high-grade ccRCC (*Figure 1G*). This clinical data aligns with our cellular experimental results, as Snail expression is known to be upregulated in high-grade ccRCC. Upon detailed comparison among grades, ABCA1 signal was more prominent in the patients with the high-grade classification (*Figure 1H*, *Figure 1—figure supplement 2*), suggesting that ABCA1 could serve as a marker for high-grade ccRCC. Xena analysis also revealed significantly higher expression of *Snail* in ccRCC, also supporting the idea that ABCA1 upregulation is correlated with exogenous Snail expression (*Figure 1—figure supplement 1C*).

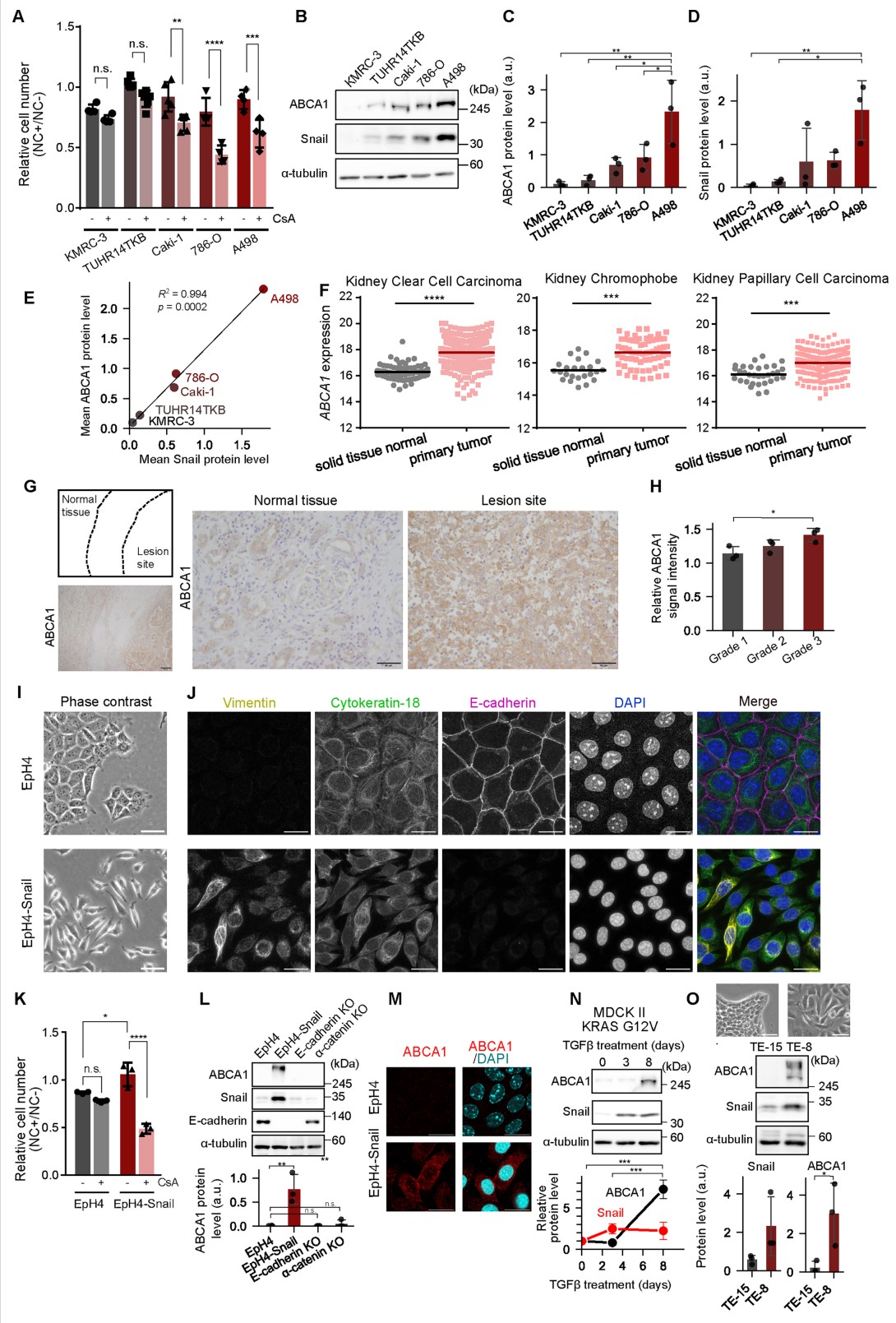

**Figure 1.** Ectopic expression of Snail confers epithelial cells nitidine chloride (NC) resistance via induction of ABCA1 expression. (**A**) Enhancement of growth inhibitory effect of NC on human renal carcinoma cells by co-treatment with an ABCA1 inhibitor cyclosporin A (CsA). Cells were grown with or without 20 μM NC and 10 μM CsA for 24 h, and cell number was determined with CCK-8. Fold change in cell number by NC was shown as relative to controls without NC. Results are shown as mean of at least four independent experiments ± SD (Tukey–Kramer's multiple comparison test). *n* = 4

*Figure 1 continued on next page*

*Figure 1 continued*

(KMRC-3,786-O,A498), 5 (Caki-1) or 6 (TUHR14TKB). (**B–E**) Immunoblot analysis of whole-cell lysate of human renal carcinoma cell lines. A representative result from three independent experiments is shown (**B**). Quantitative analyses of Snail (**C**) and ABCA1 (**D**) expression are presented. A scatter plot illustrating the correlation between ABCA1 and Snail expression levels is also shown (**E**). Quantitative data are shown as mean of three independent experiments ± SD (Tukey–Kramer's test). (**F**) Comparison of *ABCA1* expression between normal tissue and primary tumor of indicated subtypes of renal cancers analyzed using UCSC Xena (https://xenabrowser.net/). Bars indicate means (Welch's *t*-test). (**G**) Immunohistochemistry of a surgically extracted renal tissue from a patient with Fuhrman grade 3 primary ccRCC, indicating upregulation of ABCA1 in the lesion site. The ABCA1 staining was verified by the accumulation of known signals in renal tubules and glomeruli in normal tissue (*Yang et al., 2010*). Scale bars, 250 μm (left), 50 μm (right). (**H**) Quantification of ABCA1 signal from surgically extracted renal tissues from three patients for each Fuhrman grade (Tukey–Kramer's test; see also *Figure 1—figure supplement 2*). (**I**) Phase-contrast images of EpH4 wild-type cells (top) and Snail-overexpression cells (bottom, EpH4-Snail). Scale bars, 50 μm. (**J**) Immunofluorescence images of EpH4 and EpH4-Snail cells. The cells were fixed. Scale bars, 20 μm. (**K**) Acquisition of NC resistance by exogenous expression of Snail in EpH4 cells and enhancement of 20 μM NC effect on EpH4-Snail cells by co-treatment with 10 μM CsA. Data are presented as in (**A**) (*n* = 3, Tukey–Kramer's test). (**L**) Immunoblot analyses of whole-cell lysates of EpH4, EpH4-Snail, E-cadherin KO, and α-catenin KO EpH4 cells. Quantitative analyses of ABCA1 expression is also shown (*n* = 3, Dunnett"s test). (**M**) Immunofluorescence images of EpH4 and EpH4-Snail cells. Cells were fixed with MeOH. Scale bars, 20 μm. (**N, O**) Immunoblot analyses of whole-cell lysates of MDCK II cells (**N**) expressing KRAS G12V treated with 5 μg/ml TGFβ for 0, 3, and 8 days and human esophageal carcinoma cell lines TE-15 and TE-8 (**O**). Graphs on the bottom show analyses from three biological replicates (Tukey–Kramer's test [**N**] and Student's *t*-test [**O**]). Phase-contrast images are also shown on the top (**O**). Scale bars, 50 μm.

The online version of this article includes the following source data and figure supplement(s) for figure 1:

**Source data 1.** Original raw data for the immunoblot images shown in *Figure 1B, L, N and O*.

**Source data 2.** Labeled full blot images for the immunoblot images shown in *Figure 1B, L, N and O*.

**Figure supplement 1.** Cancer types that exhibit tumor-specific upregulation of ABCA1 transcription.

**Figure supplement 2.** Comparison of ABCA1 expression in different progression grades of kidney cancer cases.

Our results suggest the correlation between chemoresistance and Snail expression level. In order to clarify whether Snail expression alone can confer resistance to anti-cancer drugs in epithelial cells, Snail was ectopically expressed in normal epithelial cells, and we investigated changes in resistance to anticancer drugs. Overexpression of Snail in a mouse mammary gland epithelial cell line EpH4 abolishes transcription of various genes involved in cell–cell adhesion including E-cadherin (*Figure 1I*; *Cano et al., 2000*; *Batlle et al., 2000*; *Ikenouchi et al., 2003*). EpH4-Snail cells exhibited coexistence of the mesenchyme-specific intermediate filament Vimentin and the epithelium-specific Cytokeratin-18 (*Figure 1J*), which is a typical feature of a hybrid E/M state of cancer cells. The hybrid E/M EpH4-Snail cells were significantly more resistant to NC than parental EpH4 cells (*Figure 1K*). Consistent with the results of kidney cancer cell lines, this resistance was canceled by CsA (*Figure 1K*), and upregulation of ABCA1 expression was also observed in EpH4-Snail cells (*Figure 1L and M*). Consistent with the resistance to the antitumor agent, ABCA1 mainly localized to the plasma membrane in EpH4-Snail cells (*Figure 1M*). We also examined another model with a different cell line. In MDCK II, a canine kidney epithelial cell line, TGF-β treatment induced EMT under the expression of the oncogenic KRAS-G12V mutant (*Arner et al., 2019*). The upregulation of ABCA1 expression was also observed in this transient hybrid E/M model (*Figure 1N*), suggesting that similar mechanisms to those in EpH4-Snail are also at play in kidney cell EMT. Notably, induction of ABCA1 expression requires a longer period of treatment with TGF-β (day 8) than Snail induction (day 3), indicating that ABCA1 is upregulated in response to cellular alteration by Snail but not induced directly by TGF-β signaling. Furthermore, in comparison between human esophageal cancer cell lines (*Szlasa et al., 2020*), Snail-positive TE-8 cells exhibited higher expression of ABCA1 than Snail-negative TE-15 (*Figure 1O*). These results suggest that the induction of ABCA1 by ectopic expression of Snail occurs regardless of cellular background.

## Intracellular cholesterol accumulation drives ABCA1 expression in hybrid E/M cells

Then how does Snail induce ABCA1 expression, since as a transcriptional repressor it is incapable of doing so directly? Upregulation of ABCA1 was not observed in either E-cadherin or α-catenin KO cells (*Figure 1L*), indicating that ABCA1 induction is driven as a specific response to a fundamental change in cellular state induced by Snail overexpression, not an ancillary effect of disrupting either cell-cell adhesion or apical-basal polarity. Generally, ABCA1 is a transporter for the efflux of

cholesterol out of the cell when there is an excess of cholesterol in the cell and ABCA1 upregulation in response to excess loading of cholesterol is mediated by sterol-activated nuclear receptors, LXRs (*Ignatova et al., 2013*). Aside from the cholesterol-LXRs regulatory axis, it was recently reported that ABCA1 expression in epithelial cells is regulated via FOXO3a and c-Myc (*Frechin et al., 2015*). Activation of FAK/PI3K/Akt signaling depending on cell crowding state downregulates ABCA1 expression via phosphorylation of FOXO3a, a positive transcriptional regulator of ABCA1 (*Frechin et al., 2015*). We found that the phosphorylation level of FOXO3a at S453 tended to be lower in EpH4-Snail cells than in wild-type EpH4 cells (*Figure 2—figure supplement 1A*). However, since downregulation of Akt signaling by treatment with inhibitors of its activators GSK2334470 (PDK1), LY294002 (PI3K), and Wortmannin (PI3K) did not induce ABCA1 expression in wild-type EpH4 cells (*Figure 2—figure supplement 1A*), FOXO3a would not be responsible for ABCA1 expression in EpH4-Snail cells. Although ABCA1 expression is also downregulated by c-Myc in breast cancer, no remarkable difference in c-Myc expression was observed between EpH4 and EpH4-Snail cells, indicating that c-Myc is not likely involved in ABCA1 expression in EpH4-Snail cells (*Figure 2—figure supplement 1B*). Therefore, we next pursued in detail the possibility that LXRs induce ABCA1 in EpH4-Snail cells. Immunofluorescence microscopy revealed that LXRs are highly accumulated in the nuclei of EpH4-Snail cells, suggesting that ABCA1 upregulation could be mediated by LXRs (*Figure 2A*). Consistently, treatment with an LXR inhibitor GSK2033 abolished ABCA1 expression to an undetectable level in EpH4-Snail cells (*Figure 2B*). GSK 2033 treatment also canceled NC resistance (*Figure 2C*), suggesting that cholesterol efflux system activated by LXR is important for acquisition of chemoresistance. However, EpH4-snail cells did not show enhanced sensitivity to NC treatment even when ABCA1 was depleted by knockout (*Figure 2—figure supplement 2*). Given that CsA inhibits other isoforms of ABC transporters (*Cserepes et al., 2004*), other LXR-regulated efflux pumps may also contribute to chemoresistance.

Since transcriptional activity of LXRs is regulated by cellular cholesterol level, we examined whether ABCA1 is expressed in response to high cholesterol accumulation in the cells. Inhibition of cholesterol biosynthesis by simvastatin, an inhibitor of the rate-limiting enzyme of cholesterol biosynthesis 3-hydroxy-3-methylglutaryl-coenzyme A reductase, remarkably decreased ABCA1 expression in EpH4-Snail cells, but residual ABCA1 was still detected (*Figure 2D*, normal fetal bovine serum (FBS)). In contrast, treatment with simvastatin in the medium supplemented with lipid-free FBS completely abolished ABCA1 expression (*Figure 2D*, lipid-free FBS). Together with the finding that incubation in lipid-free FBS medium alone did not markedly decrease ABCA1 expression, these results suggest that cholesterol biosynthesis is the primary source of excess accumulation of cholesterol in EpH4-Snail cells, but cholesterol uptake from media also significantly contributes to the accumulation as a secondary source.

These findings imply dysregulation of cholesterol metabolism in EpH4-Snail. Filipin staining revealed that cholesterol was prominently accumulated in intracellular structures in EpH4-Snail cells, whereas it localized primarily in the plasma membrane of wild-type EpH4 cells (*Figure 2E*). Since excess cholesterol can be stored in lipid droplets, we stained cells with Lipi-Green to visualize lipid droplets and found that lipid droplets were significantly enlarged in EpH4-Snail (*Figure 2F and G*). Consistently, TopFluor cholesterol, a fluorescent cholesterol analogue, was localized to lipid droplets in EpH4-Snail cells (*Figure 2H*). On the other hand, the intracellular filipin signals did not co-localize with the lipid droplet protein ADRP (perilipin-2) but co-localized with the lysosomal protein LAMP1 instead (*Figure 2I*). Given that filipin stains unesterified cholesterol, internalized cholesterol accumulates in lysosomes in its unesterified form and in lipid droplets in its esterified form in EpH4-Snail cells.

We next compared the total cholesterol levels in EpH4 cells and EpH4-Snail cells; the cellular amount of cholesterol slightly increased in EpH4-Snail cells, but the difference was modest and not significant (*Figure 2J*). Therefore, the internalization of cholesterol in EpH4-Snail cells is not simply due to an increase in the amount of cholesterol, but would be due to an abnormality in the subcellular distribution of cholesterol. A large proportion (30–50%) of cellular cholesterol is associated with sphingomyelin, and depletion of sphingomyelin induces internalization of cholesterol (*Das et al., 2014*). To investigate contents of cholesterol and sphingomyelin in the plasma membrane, we utilized the lipid-binding protein probes domain 4 of perfringolysin O (cholesterol, D4) and lysenin (sphingomyelin). Intriguingly, cholesterol content in the plasma membrane was not affected by EMT, but sphingomyelin content decreased in EpH4-Snail cells (*Figure 2K and L*). Collectively, the increase in a

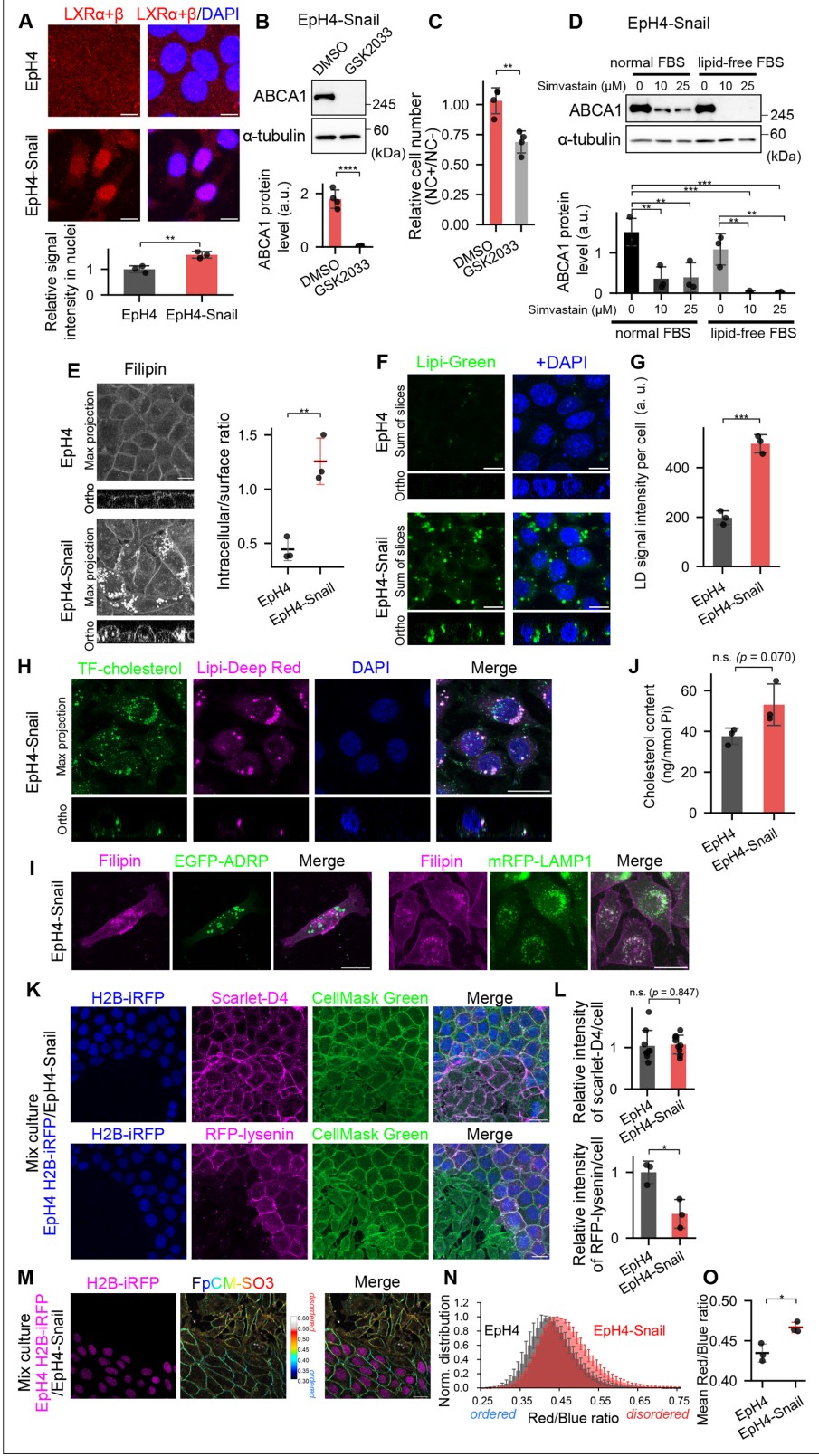

**Figure 2.** Alteration of cellular cholesterol distribution induces ABCA1 expression in hybrid E/M cells.
(**A**) Immunofluorescence images of cells stained with the LXRα+β antibody. Scale bars, 10 μm. Cells were fixed
with 4% PFA. Comparison of the abundance of LXR is also shown (bottom). LXR signal intensity overlapped with
DAPI signals was determined for three independent fields of view (Student's *t*-test). (**B**) Immunoblot analyses of

*Figure 2 continued on next page*

*Figure 2 continued*

whole-cell lysates of EpH4-Snail cells treated with 1 μM GSK-2033 for 24 h (*n*=4, Student's *t*-test). (**C**) Enhancement of growth inhibitory effect of nitidine chloride (NC) by co-treatment with an LXR inhibitor GSK2033. Cells were treated with or without 1 μM GSK2033 for 24 h, then cultured with or without 20 μM NC and 1 μM GSK2033 for 24 h, and cell number was determined with CCK-8. Fold change in cell number by NC was shown as relative to controls without NC (*n*=4, Student's *t*-test). (**D**) Immunoblot analyses of whole-cell lysates of EpH4-Snail cells treated with indicated concentration of simvastatin or 0.1% DMSO ('0' μM) in DMEM supplemented with 10% normal or lipid-free fetal bovine serum (FBS) for 48 h (*n*=3, Tukey–Kramer's test). (**E**) Subcellular distribution of cholesterol stained with filipin. Scale bar, 10 μm. Intracellular signal to surface signal ratio was determined for three independent fields of view (Student's *t*-test). (**F**) Live-cell images of lipid droplets (LDs) visualized by Lipi-Green dye. (**G**) Quantification of LD signal per cell. The Lipi-Green signals of all optical sections were summed and divided by the number of cells counted using DAPI images. Shown are means of three different fields of view ± SD. (**H**) Subcellular distribution of Topfluor-(TF-)cholesterol. Living cells were incubated with 5 μg/ml TF-cholesterol for 1 h, then stained with Lipi-Deepred and Nuc Blue. Scale bar, 20 μm. (**I**) Observation of filipin in EpH4-Snail cells expressing either EGFP-ADRP (LDs) or mRFP-LAMP1 (lysosome). Scale bar, 20 μm. (**J**) Comparison of cholesterol content relative to phospholipid (*n*=3, Student's *t*-test). (**K, L**) Comparison of cell surface cholesterol (D4) and sphingomyelin (lysenin). Wild-type EpH4 cells expressing H2B-iRFP and EpH4-Snail cells were co-cultured and stained with recombinant lipid-binding proteins and CellMask Green (**K**). Quantification of fluorescent signals of lipid binding probes (**L**). Signal intensities per cell, relative to those of wild-type cells, were quantified and analyzed (*n* = 9 (D4) or 3 (lysenin), Student's *t*-test). (**M–O**) Observation of lipid order of plasma membrane using FπCM-SO$_3$. See also *Figure 2—figure supplement 2*. Wild-type EpH4 cells expressing H2B-iRFP and EpH4-Snail cells were co-cultured and stained with 1 μM FπCM-SO$_3$ for 10 min. The hue (red/blue ratio)–brightness (intensity) images of FπCM-SO$_3$ are shown with the confocal image of H2B-iRFP (magenta) (**M**). Scale bar, 20 μm. Normalized histogram of red/blue ratio values within FπCM-SO$_3$-positive pixels (**N**). Comparison of mean red/blue ratio values (**O**) (*n*=3, Student's *t*-test).

The online version of this article includes the following source data and figure supplement(s) for figure 2:

**Source data 1.** Original raw data for the immunoblot images shown in *Figure 2B and D*, *Figure 2—figure supplement 1A and B*, and *Figure 2—figure supplement 2A*.

**Source data 2.** Labeled full blot images for the immunoblot images shown in *Figure 2B and D*, *Figure 2—figure supplement 1A and B*, and *Figure 2—figure supplement 2A*.

**Figure supplement 1.** Contribution of transcription factors involved in ABCA1 transcription in epithelial cells.

**Figure supplement 2.** Effect of ABCA1 knockout on nitidine chloride resistance of EpH4-Snail cells.

**Figure supplement 3.** Visualization and analyses of lipid order using FπCM-SO$_3$.

sphingomyelin-unbound form of cholesterol in the plasma membrane results in the internalization of cholesterol and activation of LXR to induce ABCA1 expression in hybrid E/M cells.

The strong interaction between sphingolipids and cholesterol contributes to the formation of tightly packed and less fluid plasma membrane (*van Meer et al., 2008*). Alteration of this physical property of plasma membrane frequently occurs in cancer cells and is thought to affect malignancy (*Szlasa et al., 2020*). Therefore, we next addressed the impact of sphingomyelin decrease on the lipid packing of plasma membrane using FπCM-SO$_3$, a recently developed plasma membrane-specific solvatochromic probe (*Tanaka et al., 2024*). Solvatochromic dyes respond to the polarity of their environment, exhibiting a red shift in more polar solvents. In lipid bilayers, these dyes similarly sense differences in lipid packing (order) and display a red shift in more fluid, disordered membranes (*Figure 2—figure supplement 3A*). Thus, variations in lipid order can be visualized as changes in the ratio of signal intensities of the red to blue channels (*Figure 2—figure supplement 3B*). The negatively charged and alkylated modifications enable FπCM-SO$_3$ to stably reside in the plasma membrane of living cells. Ratiometric imaging using FπCM-SO$_3$ revealed that the plasma membrane of EpH4-Snail cells was more disordered (*Figure 2M–O*). While plasma membrane fluidity is usually attributed to the cholesterol content (*Subczynski et al., 2017*), our results suggest that ectopic expression of Snail induces fluidization of plasma membrane, not by altering the surface cholesterol level but by decreasing sphingomyelin content.

## Downregulation of VLCFA-SM synthesis induces Chol/SM imbalance in hybrid E/M cells

The present findings suggest that the cause of chemoresistance may trace back to alterations of the balance of sphingomyelin and cholesterol. Therefore, we next investigate how sphingomyelin metabolism is affected by ectopic expression of Snail. We found that the total sphingomyelin content relative to total phospholipids decreased in EpH4-Snail cells (*Figure 3A*). Consequently, the ratio of cholesterol to SM (Chol/SM ratio) in the cells significantly increased in EpH4-Snail cells (*Figure 3B*). To investigate the SM profile in detail, we next analyzed the fatty acid composition of SM using LC-MS. In EpH4-Snail cells, the fractions of VLCFA-SMs (22:1, 22:0, and 24:1) were significantly decreased whereas those of LCFA-SMs (14:0, 16:1, and 16:0) were rather increased (*Figure 3C*), resulting in a prominent decrease in the total VLCFA-SM fraction (*Figure 3D*). These results suggest that a decrease in VLCFA-SMs is responsible for a decrease in total SM content in EpH4-Snail cells. On the other hand, the decrease in the cell surface content (*Figure 2L*) was more prominent than that in total SM (*Figure 3A*). Given that VLCFA-SMs are suggested to be differently trafficked in recycling from endosome to plasma membrane (*Koivusalo et al., 2007*), their reduction may lead to decreased plasma membrane sphingomyelin content by altering its intracellular distribution.

Fatty acid composition of SM is mainly modulated by substrate specificity of two families of enzymes, Elovls and CerSs, which respectively catalyze the rate-limiting step of the fatty acid elongation cycle and the N-acylation of sphingoid bases to produce ceramides (*Figure 3E*; *Levy and Futerman, 2010*; *Sassa and Kihara, 2014*). Therefore, we hypothesized that transcriptional repression of genes from these families causes the decrease in VLCFA-SM content in EpH4-Snail cells. RT-PCR analyses identified *Elovl7* and *CerS3* as genes that are repressed specifically in EpH4-Snail cells (*Figure 3F and G*). Consistent with our hypothesis, Elovl7 and CerS3 control VLCFA-SM production (*Figure 3E*). Collectively, ectopic expression of Snail represses transcription of VLCFA-SM-producing enzymes and results in a decrease in the amount of VLCFA-SM. This downregulation of biosynthesis pathway is responsible for the induction of ABCA1 expression because supplementation of VLCFA ceramide decreased the ABCA1 expression in EpH4-Snail cells (*Figure 3H*). Together with the results of the cholesterol depletion experiment (*Figure 2D*), the high Chol/SM ratio due to downregulation of VLCFA-SM biosynthesis accounts for the elevated ABCA1 expression in EpH4-Snail cells.

Since ABCA1 expression is not aligned with either the loss of epithelial state (E-cadherin and α-catenin KO; *Figure 1L*) or a nonepithelial basal state (fibroblasts), upregulation of ABCA1 is characteristic to the hybrid E/M cells expressing ectopic Snail (*Figure 3I*). To investigate the mechanism underlying this difference in ABCA1 expression, we again compared Chol/SM ratio in EpH4-Snail and normal fibroblast cell lines L-929 and NIH3T3. Chol/SM ratio of these fibroblasts was slightly but not significantly higher than that of EpH4 (*Figure 3J*). The cholesterol content in NIH-3T3 cells is significantly higher than in EpH4 cells (*Figure 3—figure supplement 1*). Consequently, normal fibroblasts would synthesize more SMs to counteract the cholesterol load imposed by the decrease in VLCFA-SMs and stabilize the Chol/SM ratio. By contrast, hybrid E/M cells are unable to respond appropriately to the harmful imbalance between cholesterol and SM due to the incomplete activation of the mesenchymal transcriptome.

To examine whether the above findings apply to human cancer cells, we investigate the Chol/SM ratio of TE-15 and TE-8 (see also *Figure 1O*; *Nishihira et al., 1993*). Consistent with our findings in the EpH4-Snail system, Chol/SM ratio was significantly higher in TE-8 cells than in TE-15 cells (*Figure 3K*). Together with the results of mouse cell lines, normal epithelial and fibroblast cells maintain Chol/SM ratio around 1.0–1.5, and ABCA1 expression as a response to the excess Chol is induced when Chol/SM ratio exceeds 1.5 (*Figure 3L*). Notably, the difference in either cholesterol or SM contents between TE-15 and TE-8 cells, and cholesterol content between EpH4 and EpH4-Snail is not significant when each is compared alone (*Figure 2D*, *Figure 3—figure supplement 1*), indicating the importance of considering their functional cooperativity, that is, to analyze the Chol/SM ratio. Collectively, the cause of chemoresistance can be traced back to the downregulation of VLCFA-SM biosynthesis (*Figure 3M*). A decrease in sphingomyelin but not in cholesterol induces plasma membrane fluidization, which leads to internalization of cholesterol. The internalized, SM-unbound cholesterol activates LXRs, which in turn elevate ABCA1 expression and thereby confer chemoresistance to cells.

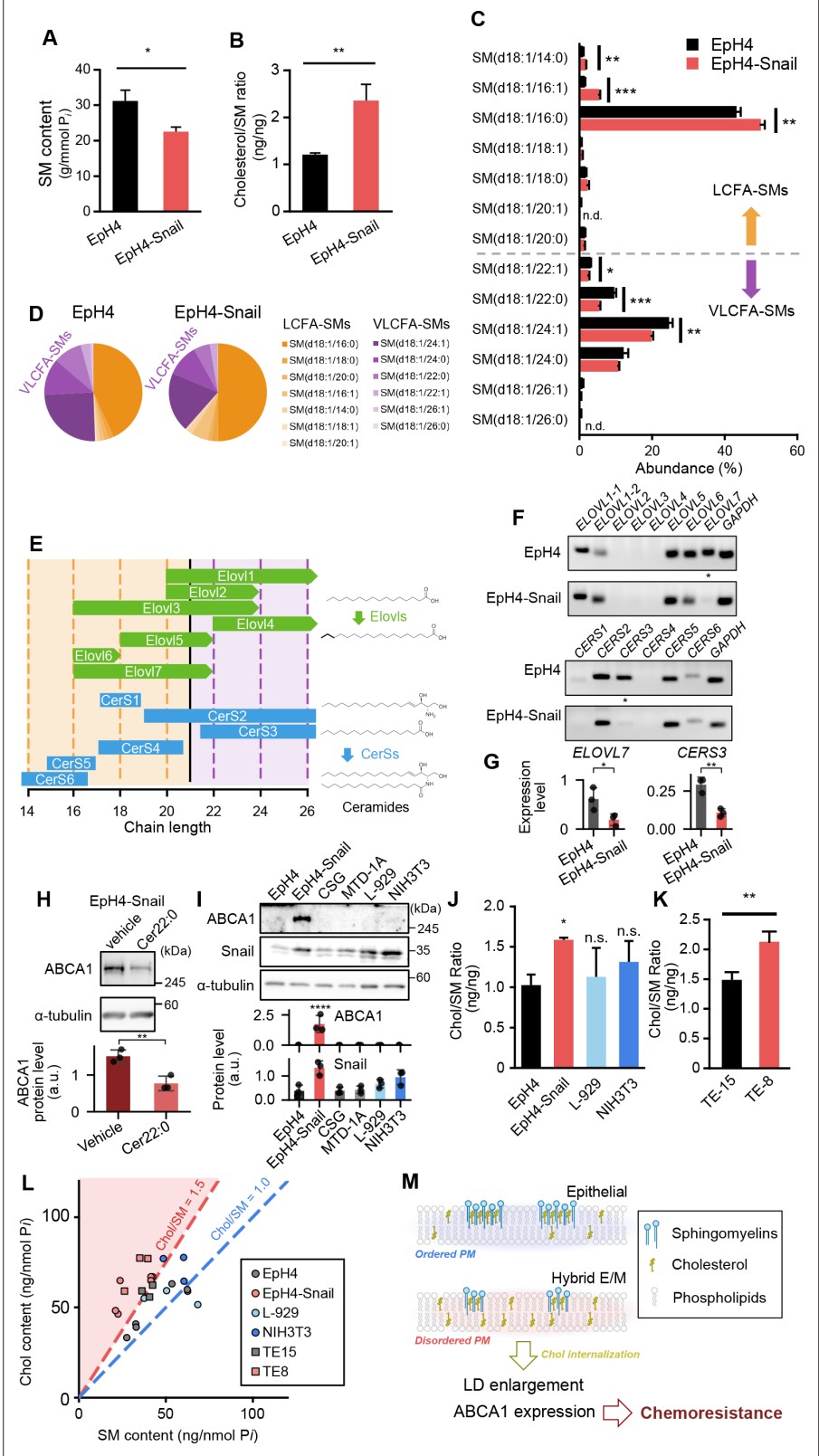

**Figure 3.** Effects of Snail-induced epithelial–mesenchymal transition (EMT) on sphingomyelin profile.
(**A**) Comparison of sphingomyelin (SM) content relative to phospholipid content (*n*=3, Student's *t*-test).
(**B**) Comparison of cholesterol (Chol)/SM ratio between EpH4 and EpH4-Snail cells (*n*=3, Student's *t*-test). (**C**) Fatty acid composition of SM determined using LC-MS. All peaks corresponding to d18:1-even fatty acid-SMs were

*Figure 3 continued on next page*

*Figure 3 continued*

included in the analysis and each peak value is expressed as a percentage. n.d., peak not detected ($n=3$, Student's *t*-test). (**D**) Pie charts of SM chain length profile from (**C**). SMs were classified into long-chain fatty acid (LCFA, orange) (≤20) and very long-chain fatty acid (VLCFA, purple) (>20) SMs. (**E**) Chain length specificity of Elovls and CerSs responsible for fatty acid profile of sphingolipids reported (**Levy and Futerman, 2010**; **Sassa and Kihara, 2014**). Elovls catalyze a rate-limiting step of fatty acid elongation cycle, and CerSs catalyze *N*-acylation of sphingoid bases to produce ceramides. (**F**) Comparison of transcription levels of *Elovls* and *CerSs*. Notable decreases in expressions of *Elovl7* and *CerS3* in EpH4-Snail cells are indicated by asterisks. (**G**) Quantification of transcription levels of *Elovl7* and *CerS3* relative to that of *ZO-1*, whose expression does not change between EpH4 and EpH4-Snail (**Ikenouchi et al., 2003**) ($n=3$, Student's *t*-test). (**H**) Effect of supplementation of C22:0 ceramide (Cer22:0) on ABCA1 expression. EpH4-Snail cells were treated with 10 µM ceramide or 0.4% vehicle (dodecane/ethanol = 2/98) for 24 h, then whole-cell lysates were analyzed by immunoblot ($n=3$, Student's *t*-test). (**I**) Immunoblot analyses of whole-cell lysates of mouse epithelial (MTD-1A, CSG) and fibroblast (L-929, NIH3T3) cells. (**J**) Comparison of Chol/SM ratio of EpH4, EpH4-Snail, and normal fibroblasts ($n=3$, Dunnett's test). (**K**) Comparison of Chol/SM ratio between TE-15 and TE-8 ($n=3$, Student's *t*-test). (**L**) Scatter plot of Chol content against SM content used in (**B**), (**H**), and (**I**). Dashed lines indicate Chol/SM = 1.0 (blue) and 1.5 (red). The region of Chol/SM >1.5 is shown with red background. (**M**) Possible mechanism of induction of ABCA1 expression and lipid droplet enlargement through decrease in VLCFA-SM. In EpH4 (epithelial) cells, a certain amount of cholesterol is sequestered through interaction with SM, organizing ordered plasma membrane (PM). In EpH4-Snail (hybrid E/M) cells, the lower level of VLCFA-SM biosynthesis leads to a decrease in plasma membrane SM content, resulting in membrane disorder. The increase in SM-unbound form of cholesterol in PM induces its internalization and activation of cellular cholesterol sensors, resulting in lipid droplet enlargement and enhanced ABCA1 expression.

The online version of this article includes the following source data and figure supplement(s) for figure 3:

**Source data 1.** Original raw data for the RT-PCR and immunoblot images shown in *Figure 3F, H and I*.

**Source data 2.** Labeled uncropped gel and full blot images for the RT-PCR and immunoblot images shown in *Figure 3F, H and I*.

**Figure supplement 1.** Comparison of SM and cholesterol content.

## Inhibitor of cholesterol esterification selectively inhibits growth of Snail-positive kidney cancer cells

Finally, we examined the possibility of targeting the imbalance of Chol/SM ratio in Snail-positive cancer cells to selectively inhibit their growth. As discussed above, hybrid E/M cells of high Chol/SM ratio must either export cholesterol out of cells or isolate it within cells in lipid droplets as cholesteryl esters to avoid the cytotoxic effects of free cholesterol. Thus, inhibiting the means of clearing excess cholesterol could specifically target cells in the hybrid E/M state for self-extermination. Cholesterol efflux from cells is mainly dependent on ABCA1, and esterification of cholesterol is catalyzed by acyl-CoA:cholesterol acyl transferases (ACATs) (*Figure 4A*; *Song et al., 2021*). Among the two ACAT isoforms in humans, SOAT1 is widely expressed, including in the kidney (*Lee et al., 1998*). Indeed, all RCC cell lines tested expressed SOAT1 (*Figure 4B*), although the expression levels did not correlate with those of Snail in the respective cell lines (*Figure 1B*). Therefore, we examined the effect of ABCA1 inhibitor CsA and ACAT inhibitor TMP-153 on the EpH4-Snail model and renal carcinoma cells. While CsA non-selectively inhibited growth of all cell lines tested (*Figure 4C and E*), TMP-153 selectively inhibited growth of Snail-positive cells (EpH4-Snail, 786-O, A498, and Caki-1; *Figure 4D and F*). Thus, sequestration of cholesterol via esterification by ACATs is the primary mechanism for handling excess cholesterol to avoid cell death in these cells. Notably, TMP-153 treatment alone did not provide this selectivity between esophageal cancer cell lines, but the combination of TMP-153 and CsA selectively inhibited growth of the Snail-positive TE-8 cells (*Figure 4—figure supplement 1A–C*). Collectively, these results suggest that inhibition of cholesterol efflux and cholesterol esterification selectively inhibits growth of cells stably in a Snail-induced hybrid E/M state.

We also investigated the potential of ACAT inhibitors as anti-tumor agents in vivo using xenograft models. We successfully transplanted the 786-O line into nude mice among Snail-positive ccRCC cell lines. Using this system, we found that TMP-153 treatment significantly reduced 786-O tumor volume in vivo (*Figure 4G–I*). Moreover, the combination of TMP-153 and CsA also inhibited the growth of TE-8 tumor xenografts (*Figure 4—figure supplement 1D–F*). Together with the in vitro assays (*Figure 4C–F*, *Figure 4—figure supplement 1A–C*), we propose that cholesterol clearance systems,

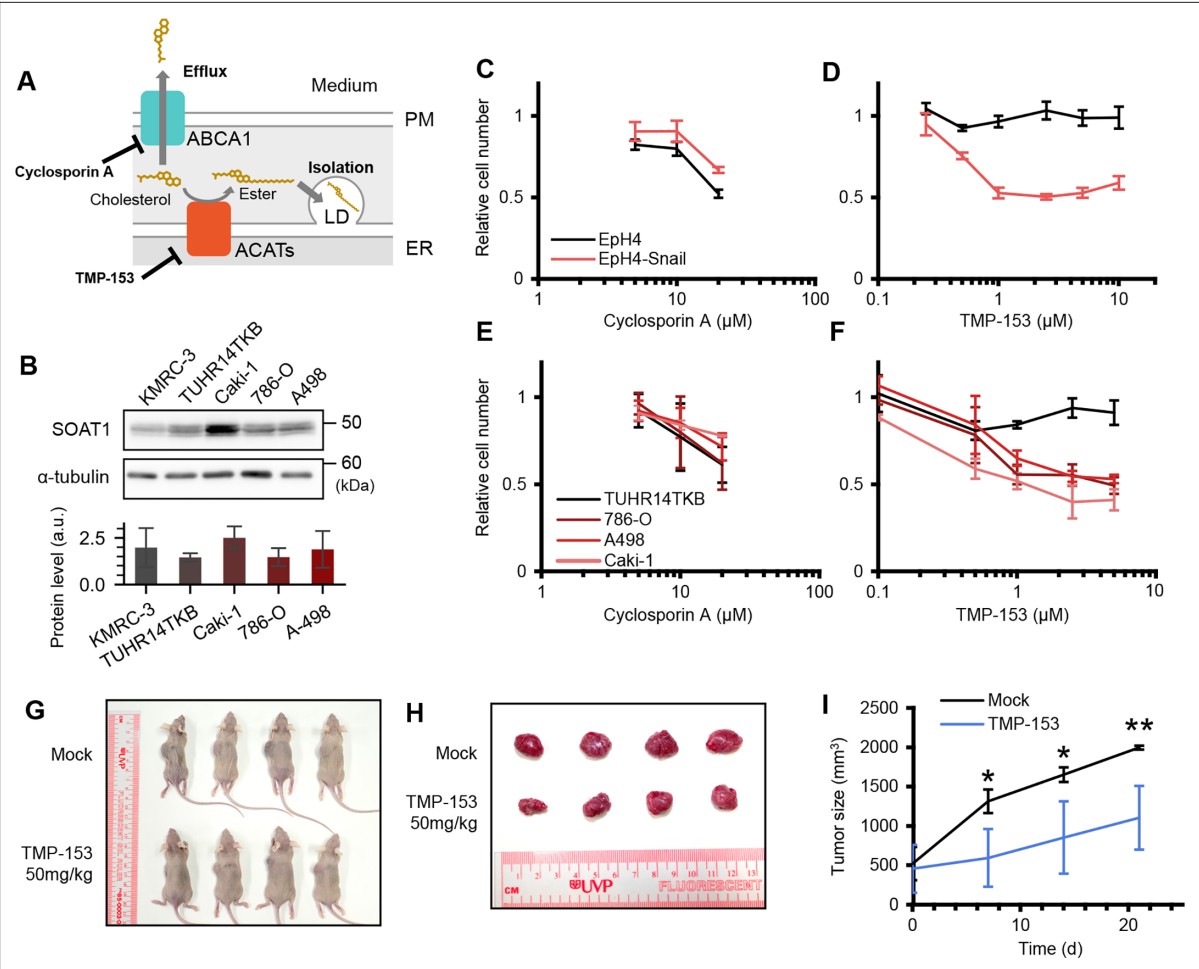

**Figure 4.** Selective growth inhibition of Snail-positive cells by an ACAT inhibitor TMP-153. (**A**) Schematic illustrating the molecular mechanisms for handling excess cholesterol to avoid cell death. Cells eliminate excess cholesterol through efflux mediated by transporters including ABCA1 or isolation into lipid droplets (LD) via esterification of cholesterol by acyl-CoA: cholesterol acyltransferases (ACATs) to avoid cell death. CsA and TMP-153 inhibit ABCA1 and ACATs, respectively. (**B**) Immunoblot analysis of whole-cell lysates of human kidney cell lines. Quantitative analysis of SOAT1 expression is also shown. (**C–F**), Effects of treatments with CsA (**C**, **E**) or TMP-153 (**D**, **F**) to cellular growth of EpH4 and EpH4-Snail (**C**, **D**) or human kidney cancer cell lines (**E**, **F**). Cells were treated with drugs for 24 h and relative cell number to 0.1% DMSO control was determined by CCK-8. Results are shown as mean of at least three independent experiments ± SD. (**G–I**) TMP-153 inhibits the growth of tumor xenografts composed of Snail-positive renal cancer 786-O cells. Experiments were performed as described in 'Materials and methods'. In brief, 50 mg/kg TMP-153 was administered intraperitoneally to nude mice bearing 786-O tumor xenografts. After 21 days of the first administration, the overall appearance (**G**) and the excised tumors (**H**) of control and treated mice were compared. (**I**) Tumor growth curve in the xenograft model. Results are shown as mean of four biological replicates ± SD (Student's *t*-test).

The online version of this article includes the following source data and figure supplement(s) for figure 4:

**Source data 1.** Original raw data for the immunoblot images shown in *Figure 4B*.

**Source data 2.** Labeled full blot images for the immunoblot images shown in *Figure 4B*.

**Figure supplement 1.** Selective growth inhibition of a Snail positive esophageal cancer cell line by combined treatment of CsA and TMP-153.

particularly ACATs, can be used as molecular targets to selectively inhibit the growth of Snail-positive cancers.

## Discussion

In recent years, it has been revealed that the expression of EMT-TFs, such as Snail, induces cancer cells to acquire mesenchymal characteristics while retaining epithelial traits, transforming them into so-called hybrid E/M cells. Hybrid E/M cells have been demonstrated to exhibit the highest plasticity

and malignancy (*Aiello et al., 2018*; *Pastushenko et al., 2018*). Moreover, Snail expression contributes to cancer stemness and resistance to chemotherapy (*Lambert and Weinberg, 2021*). To date, molecular mechanisms connecting to EMT-TFs to chemoresistance remain to be established. Here, we identify lipid metabolic imbalance-mediated ABCA1 expression as a mechanism of chemoresistance induced by Snail. We found reduced VLCFA-SM biosynthesis in Snail-positive cells with hybrid E/M phenotypes, resulting in high Chol/SM ratio. This should be detrimental in theory since it increases exposure to the toxic effect of free cholesterol, but cancer cells deploy countermeasures: upregulating ABCA1 to move cholesterol out of cells and sequestering esterified cholesterol in the form of lipid droplets. Notably, it has been also reported that not only ABCA1 (*Iwasaki et al., 2010*; *Wang et al., 2021*) but also enhanced lipid droplet formation triggers chemoresistance to various drugs (*Schlaepfer et al., 2012*; *Cotte et al., 2018*; *Hultsch et al., 2018*). Therefore, dietary or pharmacological control of cholesterol and also sphingolipids would be a new strategy to overcome resistance to chemotherapy.

The aberrant activation of EMT-TFs is implicated in multiple crucial steps in cancer progression, prompting efforts to explore therapeutic approaches aimed at suppressing EMT-TF functions in cancer cells (*Davis et al., 2014*). However, as EMT-TFs are predominantly located in the nucleus, no effective treatment methods, including antibody drugs, have been established thus far. Therefore, instead of directly targeting EMT-TFs, focusing on the metabolic network vulnerabilities specific to hybrid E/M cells may prove to be an effective therapeutic strategy. In the present study, we found increased sensitivity of Snail-positive cells to toxicity of free-cholesterol accumulation induced by the ACAT inhibitor TMP-153. Notably, kidney cancer growth could be suppressed by inhibiting cholesterol esterification alone, while esophageal cancer required the additional downregulation of ABCA1, indicating varying dependence on free cholesterol clearance strategies among cancers of different primary origin. Other cancers might employ alternative pathways to evade cytotoxic effects of excess cholesterol, and identifying these traits will pave the way for the development of novel strategies in targeted molecular therapeutics against hybrid E/M cancers. An intriguing corollary here is the potential utility of ABCA1 expression as a diagnostic marker to identify cancers that would be sensitive to targeting the Chol/SM imbalance.

Although various metabolic changes in cancer cells focusing on energy sources including sugars, amino acids, and fatty acids have been intensely studied (*Pavlova et al., 2022*), alteration of sphingolipid homeostasis driven by EMT-TFs remains largely unknown. We found both quantitative and qualitative alteration of sphingomyelin profile: decrease in the total SM amount and the VLCFA-SM percentage. This alteration is attributed to transcriptional repression of enzymes for biosynthesis of VLCFA-SMs, CerS3, and Elovl7. Genes whose transcription is repressed by Snail include those encoding proteins directly involved in epithelial adhesive structures, such as E-cadherin and Claudins. Hence, it is not surprising that VLCFA-SMs play important roles cooperatively with the adhesion molecules. Indeed, cholesterol accumulation is required for tight junction formation, and VLCFA-SMs are enriched in the junctional region (*Shigetomi et al., 2018*; *Shigetomi et al., 2023*). Studies focusing on the VLCFA-synthetic enzymes would provide a new layer of knowledge about mechanisms underlying the establishment of epithelial structures and functions. Such studies would also contribute to understanding how dysfunctions in lipid metabolism promote cancer malignancy and tissue fibrosis as a consequence of EMT. The regulation of cell adhesion by fine-tuning of lipid composition should be reversible and therefore would explain the two conflicting events: the invasion and metastasis process and the viability process at the metastatic site.

## Materials and methods
### Cells and reagents

Cells were grown in Dulbecco's Modified Eagle Medium (DMEM) supplemented with 10% FBS under 5% $CO_2$ at 37°C. Mouse Snail-expressing EpH4 cells (EpH4-Snail) (*Ikenouchi et al., 2003*), E-cadherin KO and α-catenin KO EpH4 cells (*Shigetomi et al., 2018*) were generated previously. EpH4-Snail cells were maintained in medium supplemented with 500 µg/ml G418 (Wako Pure Chemical). EpH4 cells were kindly provided by Dr. Reichmann (Institut Suisse de Recherches, Lausanne, Switzerland), and amplification of mouse-specific genomic sequences was confirmed. HEK293, 786-O, and A498 cells were obtained from the American Type Culture Collection (ATCC); Caki-1 and KMRC-3 cells were

obtained from the Japanese Collection of Research Bioresources Cell Bank; TE-8 and TE-15 cells were obtained from the Tohoku University Cell Resource Center for Biomedical Research; and TUHR14TKB cells were obtained from the RIKEN BioResource Research Center. Human cell lines were authenticated by short tandem repeat profiling using the GenePrint10 System (Promega). All cell lines were routinely tested and confirmed to be negative for mycoplasma contamination.

pLVET-HA-K-RasG12V-IRES-GFP (Addgene #107140), LAMP1-mRFP-FLAG (#34611), and pLentiPGK DEST H2B-iRFP670 (#90237) were transduced using the lentivirus produced in HEK293 cells, and fluorescence-positive cells were collected using a cell sorter Sony SH-800 as needed. pEGFP-C1-ADRP (Addgene #78161) was transfected to cells using Lipofectamine 3000 (Thermo Fisher). Nitidine chloride was purchased from Selleck Biotech, cyclosporin A from Wako, and TMP-153 from Cayman Chemical, filipin complex from *Streptomyces filipinensis* and GSK2033 from Sigma-Aldrich, Lipi-Green from Dojindo Laboratories, and ceramide (d18:1/22:0; C22:0 ceramide) from Avanti Polar Lipids.

The following primary antibodies were used for immunoblotting (IB), immunofluorescence microscopy (IF), and immunohistochemistry (IHC): rabbit anti-ABCA1 mAb (96292S, IB, IF), rabbit anti-Snail mAb (3879S, IB), rabbit anti-Akt mAb (4691S, IB), rabbit anti-phospho-Akt (T308) mAb (2965S, IB), and rabbit anti-phospho-FoxO3a (S253) pAb (9466S, IB) from Cell Signaling Technology; rabbit anti-ABCA1 pAb (NB400-105, IHC) from Novus Biologicals; rat anti-E-cadherin mAb (ECCD-2, IB, IF) from TaKaRa; mouse anti-cytokeratin 18 mAb (LS-C84878, IF) from LS Bio; goat anti-LXR alpha + LXR beta pAb (ab24362, IF) from Abcam; and mouse anti-α-tubulin mAb (12G10, IB) produced in house.

The following secondary antibodies were used: Horseradish Peroxidase (HRP)-conjugated anti-rat IgG (HAF005) and anti-mouse IgG (HAF007) from R&D Systems; HRP-conjugated anti-rabbit IgG (4030-05) from Southern Biotech; and Cy3-conjugated anti-goat IgG (705-165-147) from Jackson Immunoresearch Laboratories.

## RT-PCR

Total RNAs were prepared with RNeasy Mini Kit (QIAGEN) and the retrotranscribed with Superscript III First-Strand Synthesis System (Invitrogen). RT-PCR was then performed using KOD FX (TOYOBO) following manufacturer's instructions. Primer sequences used are shown in *Supplementary file 1*.

## Lipid analysis

Total lipids were extracted from cultured cells by Bligh–Dyer's method, dissolved in methanol/chloroform = 1/1 (v/v). Phospholipid content was determined by phosphate assay. SM and Chol content were determined using Sphingomyelin assay kit STA-601 (Cell Biolabs) and Amplex Red Cholesterol Assay Kit (Invitrogen), respectively, according to the manufacturer's instructions. Fatty acid profile of SM was determined with LC-MS as described previously (*Ikenouchi et al., 2012*). Briefly, electrospray ionization mass spectrometry analysis was performed on a 6420 triple-quadrupole liquid chromatography–mass spectrometer (Agilent Technologies) equipped with an HPLC system and an auto sampler (Infinity 1260; Agilent Technologies). The extracted lipids were directly subjected to electrospray ionization mass spectrometry analysis. The mobile phase composition was acetonitrile/methanol/water = 18:11:1 (0.1% ammonium formate). The flow rate was 4 μl/min. The mass range of the instrument was set at 650–950 *m/z*. *m/z* profiles of SM were extracted according to total ion chromatogram patterns, and all peaks corresponding to $[M+H]^+$ ions of d18:1-even chain length fatty acid-SMs were subjected to analyses.

## Fluorescence microscopy

To visualize subcellular localization of cholesterol, cells cultured on glass-bottom dishes were fixed with 4% paraformaldehyde for 15 min at room temperature (RT) and stained with 50 μg/ml filipin prepared in Phosphate-Buffered Saline (PBS) for 30 min. To visualize lipid droplets, live cells were washed once with Hank's Balanced Salt Solution (HBSS) and stained with Lipi-Green (Dojindo) for 20 min at 37°C. Cells were washed twice with HBSS and observed on a heating stage at 37°C. For immunofluorescence microscopy, cells cultured on coverslips were fixed with 4% paraformaldehyde or 2% formalin prepared in PBS for 15 min at RT and permeabilized with 0.1% Triton X-100 prepared in PBS. Fixed cells were blocked with 1% BSA prepared in PBS for 1 h at RT. Cells were incubated with primary antibodies for 1 h at RT and secondary antibodies for 1 h at RT. Antibodies were prepared in the blocking solution. For detection of ABCA1, cells were fixed with MeOH for 20 min at –30°C

and antibodies were prepared in Can Get Signal immunostain Immunoreaction Enhancer Solution B (TOYOBO). Fixed cells were observed at RT. For imaging of plasma membrane cholesterol and sphingomyelin, cells cultured on glass-bottom dishes were washed with HBSS three times, incubated with 1x working solution of CellMask Green plasma membrane stain (Molecular Probes) in HBSS, subsequently incubated with 5 µg recombinant scarlet-tagged D4 or 10 µg RFP-tagged lysenin in 100 µl HBSS for 30 min, washed three times with HBSS, and observed at 37°C. For ratiometric imaging using FπCM-SO$_3$ (*Tanaka et al., 2024*), cells cultured on glass-bottom dishes were incubated with 1 mM FπCM-SO$_3$ in HBSS for 10 min at 37°C and observed at 37°C without washout. The dye was excited with a 405 nm laser, and 400–500 nm (blue) and 500–700 nm (red) signals were split with variable dichroic mirror and simultaneously detected with two photomultipliers. The raw images were Gaussian blurred, ratio and mean intensity of two channels were calculated, and then data were expressed as a hue (ratio) – brightness (intensity) image (see *Figure 2—figure supplement 3*). All observations were performed with a confocal microscope (Carl Zeiss LSM900) equipped with Plan-APO (63×/1.40 NA, oil immersion) objective. Images were acquired using Carl Zeiss Zen 3.4 software. Images were analyzed using ImageJ/Fiji.

## Immunoblotting

Whole cells were lysed with Sodium Dodecyl Sulfate (SDS) sample buffer and samples were resolved by SDS-PAGE and transferred to nitrocellulose membranes. After blocking with 5% skim milk prepared in TBS-T, membranes were incubated with primary antibody for 1 h at RT or overnight at 4°C. Membranes were then washed and incubated with HRP-conjugated secondary antibody for 1 h at RT. Chemiluminescence signal was detected using a LAS4000mini imaging system (Fujifilm), and images were analyzed using ImageJ/Fiji. Protein levels were calculated relative to the signal intensity of α-tubulin as a loading control.

## Ceramide supplementation to culture medium

C22:0 ceramide was supplemented to culture medium as described previously (*McNally et al., 2022*). The ceramide was dissolved in dodecane/ethanol = 2/98 solution at 0.63 mM (stock solution). The stock solution (or vehicle control) was added to DMEM + 10% FBS medium at a final solvent concentration of 0.4%, resulting in a final ceramide concentration of 2.5 µM. The prepared medium was used immediately. Cells were cultured in this medium for 24 h and then subjected to further analysis.

## Growth inhibition assay

Cells cultured in 96-well microtiter plates (2×10$^3$ cells/well, 48 h) were subjected to treatments with nitidine chloride, GSK2033, CsA, TMP-153, or their combination dissolved in Dimethyl Sulfoxide (DMSO) for 24 h at 37°C. In experiments involving GSK2033, the compound was added 24 h prior to treatment with the other drugs. Final DMSO concentrations were less than 0.25%. After treatment, metabolically active cell number was determined using Cell Counting Kit-8 (Dojindo) according to the manufacturer's instructions. Briefly, 10 µl of CCK-8 solution was directly added to each well containing 100 µl culture medium and cells were incubated 1 h at 37°C. Reaction was stopped by adding 10 µl of 1% sodium dodecyl sulfate prepared in MQ and plates were put at 4°C to avoid additional reaction. Absorbance of solution at 450 nm was measured using a spectrophotometer (Thermo GENESYS 10S UV-Vis). Data were expressed as relative to 0.25% DMSO control.

## Immunohistochemistry

Formalin-fixed and paraffin-embedded surgical specimens were cut into 3 µm sections. These sections were immersed in antigen retrieval solution (pH 9.0) (Nichirei Bioscience Inc) and heated in a pressure chamber. Then, the sections were incubated with the primary antibodies (rabbit anti-ABCA1 pAb, NB400-105) and later incubated with HRP-labeled goat anti-rabbit secondary antibodies (Nichirei Bioscience Inc). Immunoreactivity was visualized by using 3–3 Diaminobenzidine Dab Substrate Kits (Nichirei Bioscience Inc). Informed consent for experimental use of the samples was obtained from all patients in accordance with the ethical guidelines of Aichi Cancer Center, and the study was approved by the Institutional Review Board (approval no. 2021-0-037). DAB signals were extracted from RGB images by color deconvolution, inverted, and quantified as the mean intensity of pixels exceeding a defined threshold using ImageJ/Fiji.

## Tumor xenograft model

Four-week-old female athymic nude mice were purchased from Japan SLC and housed in a specific pathogen-free facility. All animal experiments were conducted in accordance with the Guidelines for Animal Experiments of Aichi Medical University and approved by the Institutional Animal Care and Use Committee (approval no. 2024-58).

To initiate xenografting of 786-O, $2 \times 10^6$ cells in 200 µl DMEM were subcutaneously injected into the back of mice by using a 27G needle. After 7 days, when the inoculated tumor became visible, the mice were randomly divided into treatment and control groups, and treatment was initiated (day 0). TMP-153 (50 mg/kg body weight) or vehicle (0.5% hydroxypropyl methylcellulose with 0.1% polysorbate 80) was intraperitoneally administered to each mouse on days 0, 4, 7, 11, 14, and 18 to each mouse. The tumor volume was measured every week and calculated using the modified ellipsoid formula ($1/2 \times$ length $\times$ width$^2$). The xenografting of TE-8 was performed with some modifications. For this, $1 \times 10^7$ cells were injected. A combination of TMP-153 and CsA (both at 10 mg/kg body weight) or vehicle was intraperitoneally administered to each mouse on days 0 and 4. Finally, the tumor volume was measured 36 days after the first administration.

## Statistical analysis

Microsoft Excel for Microsoft 365 MSO 2306, GraphPad Prism v8.4.1, and Python version 3.12.7 were used for analyses and displays of quantitative data. Data are expressed as mean, with each point or error bars representing standard deviations (SD). Significant difference is indicated as follows: $*p<0.05$; $**p<0.01$; $***p<0.001$; $****p<0.0001$.

## Acknowledgements

We thank all members of the Ikenouchi Laboratory for helpful discussions, Takana Motoyoshi (KI Stainer) for immunohistochemistry, Teruaki Fujishita and Masahiro Aoki for technical advice on xenograft experiments, Minako Suzuki for technical assistance, and Kenji Kasai for pathological advice. This work was supported by AMED-FORCE (21444781) (JI), JSPS KAKENHI (JP25H00994 [JI], JP25H01325 [JI], JP25K22454 [JI], JP22KJ2374 [AM], and JP25K18465 [AM]), JST-FOREST (JPMJFR204L) (JI), and Grants from Takeda Science Foundation (JI) and the Ono Medical Research Foundation (JI).

## Additional information

### Funding

| Funder | Grant reference number | Author |
|---|---|---|
| Japan Agency for Medical Research and Development | 21444781 | Junichi Ikenouchi |
| Japan Society for the Promotion of Science | JP25H00994 | Junichi Ikenouchi |
| Japan Society for the Promotion of Science | JP25H01325 | Junichi Ikenouchi |
| Japan Society for the Promotion of Science | JP25K22454 | Junichi Ikenouchi |
| Japan Society for the Promotion of Science | JP22KJ2374 | Atsushi Matsumoto |
| Japan Society for the Promotion of Science | JP25K18465 | Atsushi Matsumoto |
| Japan Science and Technology Agency | 10.52926/JPMJFR204L | Junichi Ikenouchi |
| Takeda Science Foundation | | Junichi Ikenouchi |

| Funder | Grant reference number | Author |
|---|---|---|
| Ono Medical Research Foundation | | Junichi Ikenouchi |

The funders had no role in study design, data collection and interpretation, or the decision to submit the work for publication.

## Author contributions

Atsushi Matsumoto, Resources, Data curation, Formal analysis, Funding acquisition, Validation, Investigation, Visualization, Methodology, Writing – original draft, Writing – review and editing; Akihito Inoko, Resources, Formal analysis, Investigation, Methodology; Takuya Tanaka, Gen-Ichi Konishi, Waki Hosoda, Takahiro Kojima, Koji Ohnishi, Resources; Junichi Ikenouchi, Conceptualization, Supervision, Writing – original draft, Project administration, Writing – review and editing

## Author ORCIDs

Atsushi Matsumoto ⓘ https://orcid.org/0000-0002-3814-3299
Akihito Inoko ⓘ https://orcid.org/0000-0002-6739-2948
Junichi Ikenouchi ⓘ https://orcid.org/0000-0002-2936-3548

## Ethics

Human subjects: Informed consent for experimental use of the samples was obtained from all patients in accordance with the ethical guidelines of Aichi Cancer Center, and the study was approved by the Institutional Review Board (Approval No. 2021-0-037).
All animal experiments were conducted in accordance with the Guidelines for Animal Experiments of Aichi Medical University and approved by the Institutional Animal Care and Use Committee (Approval No. 2024-58).

Reviewer #1 (Public review): https://doi.org/10.7554/eLife.104374.3.sa1
Reviewer #2 (Public review): https://doi.org/10.7554/eLife.104374.3.sa2
Author response https://doi.org/10.7554/eLife.104374.3.sa3

# Additional files

## Supplementary files

MDAR checklist

Supplementary file 1. Primer sequences used in this study.

## Data availability

All data generated or analysed during this study are included in the manuscript and supporting files.

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
