## [Editor Report · eLife Assessment]

This article presents the **fundamental** discovery that lipid metabolic imbalance induced by Snail, an EMT-related transcription factor, contributes to the acquisition of chemoresistance in cancer cells. The evidence, supported by a wide range of methods and adequate quantification, provides a **convincing** mechanistic explanation of how Snail drives ectopic expression of the cholesterol- and drug-efflux transporter ABCA1. This work, which introduces a novel therapeutic concept targeting invasive cancer, will be of broad interest to researchers in cancer biology, lipid metabolism, and cell biology.

---

## [Referee Report · Reviewer #1 (Public review)]

The authors focus on the molecular mechanisms by which EMT cells confer resistance to cancer cells. The authors use a wide range of methods to reveal that overexpression of Snail in EMT cells induces cholesterol/sphingomyelin imbalance via transcriptional repression of biosynthetic enzymes involved in sphingomyelin synthesis. The study also revealed that ABCA1 is important for cholesterol efflux and thus for counterbalancing the excess of intracellular free cholesterol in these snail-EMT cells. Inhibition of ACAT, an enzyme catalyzing cholesterol esterification, also seems essential to inhibit the growth of snail-expressing cancer cells.

Overall, the provided data are convincing and enhance our knowledge of cancer biology.

---

## [Referee Report · Reviewer #2 (Public review)]

Summary:

This revised study provides a clearer and more mechanistically grounded explanation of how lipid metabolic imbalance contributes to EMT-associated chemoresistance in renal cancer. In this study, the authors discovered that chemoresistance in RCC cell lines correlates with the expression levels of ABCA1 and the EMT-related transcription factor Snail. They demonstrate that Snail induces ABCA1 expression and chemoresistance, and that inhibition of ABCA1-associated pathways can counteract this resistance. The study also suggests that Snail disrupts the cholesterol-sphingomyelin balance by repressing enzymes involved in VLCFA-sphingomyelin synthesis, leading to excess free cholesterol and activation of the LXR-ABCA1 axis. Importantly, inhibiting cholesterol esterification, which renders free cholesterol inert, selectively suppresses growth of a xenograft model of Snail-positive kidney cancer. These findings provide potential lipid metabolism-targeting strategies for cancer therapy. The revised version includes additional quantitative analyses and new experiments addressing lipid balance and ABCA1 localization, further strengthening the overall mechanistic model.

Strengths:

This revised manuscript provides a more comprehensive and convincing mechanistic explanation for how Snail-driven EMT induces chemoresistance through altered lipid homeostasis. The study presents a novel concept in which the Chol/SM balance, rather than individual lipid levels, shapes therapeutic vulnerability. The potential for targeting cholesterol detoxification pathways in Snail-positive cancer cells remains a significant therapeutic implication. In the revised version, the authors provide additional quantitative analyses and complementary experiments - including ABCA1 localization, restoration of VLCFA-SM levels by supplementation with C22:0 ceramide, and membrane-order assays - which further strengthen the mechanistic interpretation and address key concerns raised in earlier reviews.

Weaknesses:

The revised version includes new experiments showing that restoring sphingomyelin levels suppresses ABCA1 expression, thereby strengthening the causal link between altered lipid balance and ABCA1 induction. However, the evidence that ABCA1 is directly required for chemoresistance remains somewhat limited, as the phenotype was not reproduced by ABCA1 knockout or knockdown, and CsA may affect additional targets beyond ABCA1.

---

## [Author Response]

The following is the authors’ response to the original reviews.

**Reviewer #1 (Public review):**
The authors focus on the molecular mechanisms by which EMT cells confer resistance to cancer cells. The authors use a wide range of methods to reveal that overexpression of Snail in EMT cells induces cholesterol/sphingomyelin imbalance via transcriptional repression of biosynthetic enzymes involved in sphingomyelin synthesis. The study also revealed that ABCA1 is important for cholesterol efflux and thus for counterbalancing the excess of intracellular free cholesterol in these snail-EMT cells. Inhibition of ACAT, an enzyme catalyzing cholesterol esterification, also seems essential to inhibit the growth of snail-expressing cancer cells.However, It seems important to analyze the localization of ABCA1, as it is possible that in the event of cholesterol/sphingomyelin imbalance, for example, the intracellular trafficking of the pump may be altered.The authors should also analyze ACAT levels and/or activity in snail-EMT cells that should be increased. Overall, the provided data are important to better understand cancer biology.

We thank the reviewer for recognizing the significance of our study. Consistent with the hypothesis that ABCA1 contributes to chemoresistance in hybrid E/M cells, we agree that demonstrating the localization of ABCA1 at the plasma membrane is important, and we have included additional experiments to address this point.

We also examined the expression of the major ACAT isoform in the kidney, SOAT1, across RCC cell lines. However, its expression did not correlate with that of Snail (Figure 4B), suggesting that SOAT1 is constitutively expressed at a certain level regardless of Snail expression. The details of these additional experiments are provided in the point-by-point responses below.

**Reviewer #2 (Public review):**
Summary:In this study, the authors discovered that the chemoresistance in RCC cell lines correlates with the expression levels of the drug transporter ABCA1 and the EMT-related transcription factor Snail. They demonstrate that Snail induces ABCA1 expression and chemoresistance, and that ABCA1 inhibitors can counteract this resistance. The study also suggests that Snail disrupts the cholesterol-sphingomyelin (Chol/SM) balance by repressing the expression of enzymes involved in very long-chain fatty acid-sphingomyelin synthesis, leading to excess free cholesterol. This imbalance activates the cholesterol-LXR pathway, inducing ABCA1 expression. Moreover, inhibiting cholesterol esterification suppresses Snail-positive cancer cell growth, providing potential lipid-targeting strategies for invasive cancer therapy.Strengths:This research presents a novel mechanism by which the EMT-related transcription factor Snail confers drug resistance by altering the Chol/SM balance, introducing a previously unrecognized role of lipid metabolism in the chemoresistance of cancer cells. The focus on lipid balance, rather than individual lipid levels, is a particularly insightful approach. The potential for targeting cholesterol detoxification pathways in Snail-positive cancer cells is also a significant therapeutic implication.Weaknesses:The study's claim that Snail-induced ABCA1 is crucial for chemoresistance relies only on pharmacological inhibition of ABCA1, lacking additional validation. The causal relationship between the disrupted Chol/SM balance and ABCA1 expression or chemoresistance is not directly supported by data. Some data lack quantitative analysis.

We thank the reviewer for his/her insightful and constructive comments. In response, we have performed additional experiments using complementary approaches to further substantiate the contribution of Snail-induced ABCA1 expression to chemoresistance. Furthermore, to clarify the causal relationship between reduced sphingomyelin biosynthesis and ABCA1 expression, we conducted new experiments showing that supplementation with sphingolipids attenuates ABCA1 upregulation (Figure 3H). The details of these additional experiments are described in the point-by-point responses below.

**Reviewer #1 (Recommendations for the authors):**
In this paper, the authors reveal that snail expression in EMT-cells leads to an imbalance between cholesterol and sphingomyelin via a transcriptional repression of enzymes involved in the biosynthesis of sphingomyelin.This paper is interesting and highlights how the imbalance of lipids would impact chemotherapy resistance. However, I have a few comments.In Figure 2 in Eph4 cells, while filipin staining appears exclusively at the plasma membrane in the case of EpH4-snail cells filipin staining is also intracellular. It seems plausible that all filipin-positive intracellular staining is not exclusively in LDs, authors should therefore try to colocalize filipin with other intracellular markers. To this aim, authors might want to use topfluocholesterol-probe for instance.

We examined the distribution of TopFluor-cholesterol in hybrid E/M cells (Figure 2H) and found that TopFluor-cholesterol colocalizes with lipid droplets. In addition, we analyzed the colocalization between intracellular filipin signals and organelle-specific proteins, ADRP (lipid droplets) and LAMP1 (lysosomes) (Figure 2I). Since filipin binds exclusively to unesterified cholesterol, filipin signals did not colocalize with ADRP. Instead, we observed colocalization of filipin with LAMP1, suggesting that cholesterol accumulates in hybrid E/M cells in both esterified and unesterified forms.

In Figure 3, the authors reveal that the exogenous expression of the snail alters the ratio of cholesterol to sphingomyelin. The authors should reveal where is found the intracellular cholesterol and intracellular sphingomyelin within these cells Eph4-snail.

To investigate the lipid composition of the plasma membrane, we utilized lipid-binding protein probes, D4 (for cholesterol) and lysenin (for sphingomyelin) (Figures 2L and 2M). We found that the plasma membrane cholesterol content was not affected by EMT, whereas sphingomyelin levels were markedly decreased. In addition, intracellular cholesterol was visualized (Comment 1-1; Figures 2E–2K). On the other hand, because visualization of intracellular sphingomyelin is technically challenging, we were unable to include this analysis in the present study. We consider this an important direction for future investigation.

Regarding the model described in panel K of Figure 3. I would expect that the changes in lipid-membrane organization depicted in panel K should affect the pattern of GM1 toxin for instance or the motility of raft-associated proteins for instance. The authors could perform these experiments in order to sustain the change of lipid plasma membrane organization.

We attempted staining with FITC–cholera toxin to visualize GM1, but both EpH4 and EpH4–Snail cells exhibited very low levels of GM1, resulting in minimal or no detectable staining (data not shown). Instead, to assess the impact of decreased sphingomyelin on the overall biophysical properties of the plasma membrane, we used a plasma membrane–specific lipid-order probe, FπCM–SO₃ (Figures 2N–2P and Figure 2—figure supplement 3). We found that the plasma membrane of EpH4–Snail cells was more disordered (fluidized), suggesting that the overall properties of the plasma membrane are altered by ectopic expression of Snail.

Another issue is the intracellular localization of ABCA1 in Eph4-Snail cells. Knowing that a change in the cholesterol/sphingomyelin ratio can also modify intracellular protein trafficking, it seems important to analyze the intracellular localization of ABCA1 in EPh4-Snail cells.

We performed immunofluorescence microscopy for ABCA1 and found that ABCA1 was mainly localized at the plasma membrane in EpH4–Snail cells (Figure 1M).

As for the data on ACAT inhibition, we expect an increase in ACAT activity and protein levels in EMT cells overexpressing Snail. The authors should also investigate this point.

As noted in our response to the public review, we examined the expression of the major ACAT isoform in the kidney, SOAT1, across RCC cell lines. However, its expression did not correlate with Snail (Figure 4B), suggesting that SOAT1 is expressed at sufficient levels even in cells with low Snail expression. We agree that measuring ACAT activity would be important, as ACATs are regulated at multiple levels. However, we consider this to be beyond the scope of the present study and plan to address it in future work.

Minor commentsI do not understand why in the text, Figure S1 appears after Figure S2. The authors might want to change the numbering of these two figures.

We thank the reviewer for pointing this out. We have corrected the numbering of the supplementary figures so that Figure S1 now appears before Figure S2 in both the text and the revised figure legends.

Page 5, lane 20 Figure 1I instead of 1H.Page 6, lane 2, Figure 1J instead of 1I, and lane 9 Figure 1H instead of 1I.

We thank the reviewer for carefully checking the figure references. We have corrected the figure numbering errors in the text as suggested.

**Reviewer #2 (Recommendations for the authors):**
For Figures 1B, 1H, 1J, 2B, 2C, 3G, S3A, and S3B, to enhance data reliability, it is necessary to conduct a quantitative analysis of the Western blot data. The average values from at least three biological replicates should be calculated, with statistical significance assessed.

We have conducted quantitative analyses of the Western blot data for Figures 1B, 1H, 1J, 2B, 2C, 3G, S3A, and S3B. Band intensities from at least three independent biological replicates were quantified, and the mean values with statistical significance are now presented in the revised figures.

For Figures 1D, 2A, 2D, and S2, the images of cells or tissues should not rely solely on selected fields. Quantitative analysis is required, and the mean values from at least three biological replicates should be provided with statistical significance testing.

We have performed quantitative analyses for Figures 1D, 2A, 2D, and S2. The quantification was based on data from at least three independent biological replicates, and the mean values with statistical significance are now included in the revised figures.

For Figures 1A, 1G, 4, and S5, evaluating ABCA1's involvement in drug resistance based solely on CsA treatment is insufficient. Demonstrating the loss of drug resistance through ABCA1 knockdown or knockout is necessary.

We generated ABCA1 knockout EpH4–Snail cells and examined their resistance to nitidine chloride. However, knockout of ABCA1 alone did not affect resistance to the compound (Figure 2 - figure supplement 2). This may be due to secondary metabolic alterations induced by ABCA1 loss or compensatory upregulation of other LXR-induced cholesterol efflux transporters. Instead, we demonstrated that treatment with the LXR inhibitor GSK2033 reduced the nitidine chloride resistance of EpH4–Snail cells (Figure 2C), supporting the idea that enhanced efflux of antitumor agents through the LXR–ABCA1–mediated cholesterol efflux pathway contributes to nitidine chloride resistance.

For Figure 3, to establish a causal relationship between changes in the Chol/SM balance and ABCA1 expression, it is important to test whether modifying cholesterol and SM levels to disrupt this balance affects ABCA1 expression.

Regarding causality, as shown in Figure 2, we have already demonstrated that reducing cholesterol levels in EpH4–Snail cells decreases ABCA1 expression. To further explore this relationship, we examined whether increasing sphingomyelin levels by adding ceramide to the culture medium—thereby restoring the sphingomyelin-to-cholesterol ratio—would reduce ABCA1 expression (Figure 3H). Indeed, supplementation with C22:0 ceramide decreased ABCA1 expression, suggesting that downregulation of the VLCFA-sphingomyelin biosynthetic pathway triggers ABCA1 upregulation. Collectively, these findings support a causal relationship between the Chol/SM balance and ABCA1 expression.

In Figure 3, if there is any information on differences in cholesterol affinity between LCFA-SM and VLCFA-SM, it would be beneficial to include it in the manuscript.

Differences in cholesterol affinity between LCFA-SM and VLCFA-SM in cellular membranes remain controversial and have yet to be fully elucidated. The decrease in cell surface sphingomyelin content, evaluated by lysenin staining (Figure 2L), was more pronounced than that of total sphingomyelin (Figure 3A). Given that VLCFA-SMs have been suggested to undergo distinct trafficking during recycling from endosomes to the plasma membrane (Koivusalo et al. Mol Biol Cell 2007), their reduction may lead to decreased plasma membrane sphingomyelin content by altering its intracellular distribution. We have added this discussion to the revised manuscript.

In Figure 3F, it is recommended to assess housekeeping gene expression as a control. Quantitative real-time PCR should be performed, and the average values from at least three biological replicates should be presented.

We have performed quantitative RT-PCR analysis. The average values from at least three independent biological replicates are presented in Figure 3G.

For Figure 3F, to show whether the reduction of CERS3 or ELOVL7 affects the Chol/SM balance and ABCA1 expression, it is necessary to investigate the phenotypes following the knockdown or knockout of these enzymes.

We fully agree that phenotypic analyses of epithelial cells lacking CerS3 or ELOVL7 would provide valuable insights. However, we consider such investigations to be beyond the scope of the present study and plan to pursue them in future work.

Clarifying whether similar phenotypes are induced by other EMT-related transcription factors, or if they are specific to Snail, would be beneficial.

We agree that examining whether similar phenotypes are induced by other EMT-related transcription factors would be highly valuable for understanding the broader EMT network. However, as the focus of the present study is on lipid metabolic alterations associated with EMT—particularly the imbalance between sphingomyelin and cholesterol—we consider this investigation to be beyond the scope of the current work and plan to address it in future studies.

There are errors in figure citations within the text that need correction:p.9 l.18 Fig. 3D → Fig. 3Gp.9 l.22 Fig. 3I → Fig. 3Hp.9 l.23 Fig. S2 → Fig. S4p.10 l.6 Fig. 3J → Fig. 1Jp.10 l.8 Fig. 3J → Fig. 1Jp.10 l.9 Fig. 3K → Fig. 3Ip.10 l.12 Fig. 3H → Fig. 3Jp.10 l.14 Fig. 2D and Fig. S4 → Fig. 2G and Fig. S4D

We thank the reviewer for carefully pointing out these citation errors. We have corrected all figure references in the text as suggested.